

# Maternal self-efficacy and emotional well-being in Chilean adolescent mothers: the relationship with their children's social-emotional development

Laura Léniz-Maturana[1], Rosa Vilaseca[1] and David Leiva[2]

[1] Department of Cognition, Development and Educational Psychology, Universitat de Barcelona, Barcelona, Spain
[2] Department of Social Psychology and Quantitative Psychology, Universitat de Barcelona, Barcelona, Spain

Corresponding author
Laura Léniz-Maturana,
laura.leniz.maturana@gmail.com

## ABSTRACT

**Background:** Low maternal self-efficacy and high levels of anxiety, depression, and stress can be triggered in adolescent mothers due to an incomplete development process that makes them physically or psychologically unprepared for the responsibilities of motherhood and parenting. These factors may be linked to difficulties with their children's social-emotional development. The present study aims to: (a) analyze the relationship between maternal self-efficacy and stress, depression, and anxiety levels in low-income adolescent mothers; (b) examine the relationship between maternal self-efficacy and well-being with children's social-emotional development; and (c) describe the effects of maternal self-efficacy on children's social-emotional development, mediated by maternal well-being.

**Methods:** A sample of 79 dyads comprising low-income Chilean adolescent mothers aged from 15 to 21 years old (M = 19.1, SD = 1.66) and their children aged 10 to 24 months (M = 15.5, SD = 4.2) participated in this research. A set of psychometric scales was used to measure maternal self-efficacy (Parental Evaluation Scale, EEP), the mothers' anxiety and depression (Hospital Anxiety and Depression Scale, HADS), maternal stress (Parental Stress Scale, PSS), and the children's social-emotional development (Ages and Stages Questionnaire Socio-emotional, ASQ-SE). Bivariate analyses and mediation models were employed to estimate and test the relevant relationships.

**Results:** A bivariate analysis showed that maternal self-efficacy was negatively related to the mother's anxiety, depression, and stress. Moreover, there was a significant relationship between maternal self-efficacy and maternal stress, and children's self-regulation and social-emotional development. Maternal self-efficacy, mediated by maternal anxiety, depression, and stress scores, had a significant effect on the development of children's self-regulation.

**Conclusions:** The results confirm the importance of adolescent mothers' emotional well-being and maternal self- efficacy with respect to their children's social-emotional development. This makes it necessary to have detailed information about how emotional and self-perception status influences a mother's role in the development of her children.

## INTRODUCTION

Research into the influence of parents' emotional well-being and self-efficacy on children's social-emotional development has recently been subject to growing attention (*Davis et al., 2020*; *Lawrence et al., 2020*; *Maria et al., 2020*). However, the conditions of emotional distress experienced by parents vary depending on the circumstances, one of which is early motherhood. Infants born to adolescent mothers could be at higher risk of physical, cognitive, and social-emotional problems, given that these mothers are more prone to complications during pregnancy and emotional problems. Risk factors identified in adolescent mothers include a lower income level, a higher likelihood of school dropout and lower maternal self-efficacy (*Uzun et al., 2013*).

In Chile, the probability of becoming pregnant during adolescence has traditionally been higher among women with low socioeconomic status; therefore, many young pregnant females from vulnerable sectors discontinue their studies (*Rodríguez-Vignoli, 2005*). In recent decades, the school dropout rate among low-income youth has been higher among mothers (*Olavarría & Molina, 2012*). It should be noted that the number of adolescent mothers has decreased over the last decade (*Instituto Nacional de Estadística, 2017*). According to the National Institute of Statistics of Chile, however, 7.9% of births occurred to adolescent mothers, which is still a significant percentage of the population, and it is more likely to occur in adolescents with low education levels than in females of the same age but with higher education levels (*Rodríguez-Vignoli & San Juan Bernuy, 2020*). Likewise, the research carried out by *Berthelon & Kruger (2017)* showed that adolescent motherhood in a Chilean sample had adverse effects on educational outcomes (*i.e.*, high school completion and enrolment in technical or university studies). Consequently, labor insertion is more difficult for these women than for those who did not become mothers during adolescence. Thus, this situation perpetuates the poverty of adolescent mothers and their children.

Although Chile has a lower rate of adolescent mothers than other Latin American countries, the characteristics of adolescents who are more likely to become pregnant at an early age are similar (for instance, low economic and educational levels) (*CEPAL, 2018*). However, unlike other Western countries such as Mexico and the United States, in Chile adolescent pregnancies are not related to ethnicity or religion (*Huneeus et al., 2018*; *Jiskrova & Vazsonyi, 2019*; *Mejía-Benavides et al., 2019*). Likewise, along with England and Ethiopia, Chile stands out in terms of the measures that the government has taken to reduce the rate of pregnancy in adolescence (*Chandra-Mouli et al., 2019*). Furthermore, compared to other Latin American countries, Chile is relatively prosperous, although the economic differences inside the country are greater than in other OECD countries (*Banco Mundial, 2020*). Therefore, adolescent mothers continue to belong to the lowest socioeconomic levels, and some are likely to have more than one child (*Instituto Nacional de Estadística, 2019*). A study by *Luttges et al. (2021)* found that Chilean adolescent
mothers require support in parenting strategies, since they lack the skills needed to cope with the difficulties of motherhood (*Erfina et al., 2019*; *Mangeli et al., 2017*). Scientific evidence has shown that child-rearing conflict in adolescent mothers and economic problems, such as those experienced by Chilean adolescent mothers, are associated with parental distress and self-efficacy, which are, in turn, related to problems with children's social-emotional development (*Sanglee et al., 2019*). Few studies have focused on the relationship between the variables associated with the emotional well-being and self-efficacy of adolescent mothers and the social-emotional development of infants, particularly in Chilean dyads. So, it is important to consider the children's social-emotional development when analyzing maternal factors in adolescents, and to bear in mind the unfavorable environment in which they live, characterized by poverty, lack of support, and drop-out from school.

## The concept of maternal self-efficacy

The traditional definition of self-efficacy in the literature is based on Bandura's theory *Bandura (1989*, *1997)*, which characterizes it as a person's belief or confidence in their ability to perform a specific task and to handle obstacles in a way that allows them to successfully fulfil their expectations. Specifically with regard to parenting, the author defines it as the belief of parents in their ability to overcome the difficulties that may arise and to promote their children's development and positive adaptation. More recently, parenting self-efficacy was defined by *Kendall & Bloomfield (2005)* as a parent's own perception of their ability to successfully care for their children. Other researchers, such as *Spielman & Taubman-Ben-Ari (2009)*, have proposed that parental self-efficacy consists of a parent's behavior regarding parenting responsibilities and their degree of confidence to fulfill the parental role. In this regard, parental self-efficacy refers to personal beliefs about one's own ability to be a "good father or mother." Thus, parents who believe in their own abilities generally feel more satisfied and able to do what is necessary to persevere and thus achieve a specific task (*Farkas-Klein, 2008*). Parents who view themselves as competent feel safer in their attitudes and behaviors and are more capable of solving problems (*Coleman & Karraker, 1998*).

However, it is important to keep in mind that parenting is a responsibility that requires parents to adapt their abilities to respond to changing demands, and that these abilities are necessary for children to develop appropriately (*Garay-Gordovil, 2013*). Specifically, the quality of relationships with primary caregivers in the first years of life is crucial for children's development (*Thompson, 2001*), and parental self-efficacy is an influential factor in the quality of care they provide (*Sanders & Woolley, 2005*). Therefore, it is essential to understand parental self-efficacy during this stage, because children depend on their parents' care, which shapes their social and emotional adjustment (*Weaver et al., 2008*). There is a relationship between the parents' sense of effectiveness and their commitment to applying parenting strategies that promote the proper care of their children, thereby resulting in good learning and well-being outcomes for their children (*Ardelt & Eccles, 2001*).

Unlike fathers, mothers usually experience biological changes which can generate psychological problems and put them at an increased risk of postnatal depression (*Teti & Gelfand, 1991*). These alterations may affect maternal self-efficacy (*Farkas-Klein, 2008*). Furthermore, a mother's experience of childbirth (*Byrne et al., 2014*), the support she receives, and the characteristics of the child (*Weaver et al., 2008*) are associated with maternal self-efficacy (*Leahy-Warren, Mccarthy & Corcoran, 2012*). It has also been reported that children of mothers with high maternal self-efficacy cry less than children of mothers with low self-efficacy (*Bolten, Fink & Stadler, 2012*). When women do not perceive themselves as efficient mothers, this may cause them emotional distress, which could interfere in their child's social-emotional development (*Huhtala et al., 2014*).

## The association between maternal self-efficacy and maternal well-being

Emotional well-being is a concept that has been addressed in numerous fields of research. In this study, it is aligned with the concept of psychological well-being, which is associated with mental health and quality of life. In this context, an absence of mental illness improves quality of life and promotes better emotional well-being (*Schrank et al., 2013*). By contrast, anxiety, depression, and stress are risk factors for emotional well-being. The concept of anxiety refers to an intense and complex emotional state that includes feelings of tension and apprehension. It varies depending on the stress experienced by a person, affects the mental state, and is involved in the activation of the nervous system (*Spielberger & Reheiser, 2009*).

Depression is defined as a mental disorder associated with emotions of sadness and grief that persist over time, even when the external cause of these sensations has disappeared, are disproportionate to their cause and result in a loss of interest and pleasure (*World Health Organization, 2017*). Stress is understood as reactions associated with different problems that involve physiological, cognitive, emotional, behavioral, and sociocultural components and that require the individual to reformulate their lifestyle (*Selye, 1976*). A considerable amount of research has reported that higher levels of depression, anxiety, and stress are associated with lower maternal self-efficacy (*Atkins, 2010*; *Haslam, Pakenham & Smith, 2006*; *Kohlhoff & Barnett, 2013*; *Leahy-Warren, Mccarthy & Corcoran, 2012*; *Tahmassian & Anari, 2011*). More specifically, maternal self-efficacy can be a protective factor against maternal distress (*Law et al., 2019*; *Monteiro et al., 2020*).

Different factors could generate low maternal self-efficacy and social-emotional problems in mothers, including poor social support. For example, adolescent mothers often lack the necessary strategies to fulfill the responsibilities of motherhood (*Erfina et al., 2019*; *Mangeli et al., 2017*). However, when they receive family support, their maternal self-efficacy is higher (*Puspasari, Nur Rachmawati & Budiati, 2018*). Furthermore, it is well known that the characteristics of the child and the mother influence motherhood, and some factors may cause distress for mothers, thereby generating adverse effects on motherhood and the child's development (*Siller & Sigman, 2008*; *Warren et al., 2010*). Specifically, problems associated with emotional well-being may affect parents' conduct, thus making their children more likely to engage in aggressive and dissocial behaviors and

to suffer from anxiety, depression, social isolation, and low self-esteem (*Gracia, Lila & Musitu Ochoa, 2005*). From this standpoint, it is essential to recognize the role played by maternal self-efficacy in the well-being of mothers and their children (*Albanese, Russo & Geller, 2019*). The literature suggests that maternal self-efficacy may have a direct influence on children's social interaction and self-regulation. However, it could also be mediated through other parental factors (*Jones & Prinz, 2005*), such as high levels of anxiety and depression in mothers. Furthermore, maternal stress affects childhood behavior, increasing externalizing and internalizing problems (*Chung et al., 2013*; *Winstone, Curci & Crnic, 2020*).

It has been shown that proposing strategies designed to improve adolescent mothers' ability to care for their children helps to increase levels of maternal self-efficacy and in turn reduces levels of emotional distress (*Mohammad et al., 2021*) It is known that low levels of maternal self-efficacy in adolescent mothers can affect their confidence in their capacity to fulfill their responsibilities with regard to their child (*Lara et al., 2017*), and that this in turn may well affect their child's social-emotional development (*Agnafors et al., 2019*).

## The relationship between maternal self-efficacy and well-being during adolescence and children's social-emotional development

Adolescence is a complicated stage of development, due to the sudden physical changes and the beginning of a process of individuation in response to social demands (*Madruga & Queija, 2010*). The physical and emotional changes in adolescents generate specific age-dependent needs (*Borrás, 2014*). Research by *Colarossi & Eccles (2003)*, *Oliva (2006)*, *Shochet et al. (2006)*, *Wight, Botticello & Aneshensel (2006)*, and *Wille, Bettge & Ravens-Sieberer (2008)* has shown that young people who feel accepted and included by their friends and parents and receive support and kindness develop better individual skills, which reduces the likelihood of mental health problems. Likewise, these authors indicated that low income is a risk factor for anxiety and depression in adolescents. The symptoms mentioned above may be triggered in adolescent mothers because their development process is incomplete; they undergo major biological changes, marked by physical growth and sexual maturation, and they may not be physically or psychologically prepared to take on the responsibilities of motherhood (*Aracena et al., 2005*).Maternity in adolescence is generally associated with poverty (*Berry et al., 2000*), and social isolation and low maternal self-efficacy are variables associated with high levels of depression (*Birkeland, Thompson & Phares, 2005*). In turn, depression and stress in adolescent mothers have been associated with developmental delays in their infants (*Huang et al., 2014*).

Thus, pregnancy has unfavorable consequences for the daily lives of adolescents. One of the most common is school dropout (*Gyan, 2013*), which could affect their emotional well-being, thereby impairing their strategies to tackle parenting challenges and interfering with their perception of their capacity to successfully fulfill maternal responsibilities (*Mangeli et al., 2017*). In addition, suicide attempts, higher levels of depression, and anxiety are more likely to occur among pregnant adolescents than among their

non-pregnant peers. Moreover, adolescents, in turn, are more likely to become pregnant if a parent has died during childhood, or if there has been alcohol or drug use, a suicide attempt, the suicide of a relative, threats of physical or sexual abuse, and poor social and psychosocial support in the family (*Bonilla-Sepúlveda, 2010*; *Freitas et al., 2008*).

A parent's emotional well-being can be affected by age, marital status and a lack of strategies to cope with difficulties and problems in their child's behavior (*Abidin, 1992*; *Reid & Meadows-Oliver, 2007*), factors that can cause anxiety and stress. In that context, it is well known that mothers who have low self-esteem during pregnancy may suffer from postpartum depression and suicidal thoughts (*Dinwiddie, Schillerstrom & Schillerstrom, 2018*; *Islam et al., 2020*). Stressors related to depression in adolescent mothers are associated with low maternal self-efficacy (*Lara et al., 2017*). However, when adolescent mothers have family support, their mental health is better (*Angley et al., 2015*). From this perspective, the psychological and social environment of adolescent mothers plays a key role in their well-being and maternal self-efficacy, including family variables, education level, socioeconomic level, and the father's degree of commitment to the child.

Scientific evidence indicates that children's social-emotional development, including emotional and behavioral problems, is affected when their mothers have high levels of anxiety and depression during the prenatal and postnatal period (*Glasheen, Richardson & Fabio, 2010*; *O'Connor et al., 2002*; *Sohr-Preston & Scaramella, 2006*). The foundations of an adequate social-emotional development are established during the first years of life and depend to a great extent on the care environment (*Humphreys, Zeanah & Scheeringa, 2005*). From this point of view, the quality of the relationship between caregivers and children is fundamental for promoting better social-emotional development over time (*Scherer et al., 2019*; *Weed, Keogh & Borkowski, 2006*).

Several studies agree that the children of adolescent mothers develop more slowly than those of non-adolescent mothers, and one of the possible causes of these outcomes is the socially disadvantaged origin of these younger mothers (*Huang et al., 2014*; *Mollborn & Dennis, 2012*; *Morinis, Carson & Quigley, 2013*). Notably, children of adolescent mothers have more significant behavioral problems, lower levels of social performance and poorer well-being (*Hofferth & Reid, 2002*; *Levine, Pollack & Comfort, 2001*). In addition, it is well known that high-quality interactions between adolescent mothers and their children is related to positive social-emotional development (*Williams, 2020*). However, children who generally exhibit more externalizing problems and show lower adaptive behavior are usually associated with high neglect potential in mothers (*Lounds, Borkowski & Whitman, 2006*).

Following on from the references mentioned in the introduction, the overall aim of the current study is to examine the relation between maternal self-efficacy and well-being, and social-emotional development in children born to Chilean adolescent mothers. Specifically we aim to: (a) analyze the relationship between maternal self-efficacy and emotional well-being (*i.e.*, levels of stress, depression, and anxiety) in adolescent mothers; (b) examine the relationship between maternal self-efficacy and well-being with children's social-emotional development; (c) describe how maternal well-being mediates the effects of maternal self-efficacy on children's social-emotional development and its areas:

self-regulation, adaptive-functioning, affect, social-communication and interaction through mediation models (one for each social-emotional area). On the basis of the evidence reviewed above, we propose three hypotheses:

H.1 Higher maternal self-efficacy levels in adolescent mothers are associated with higher maternal emotional well-being. Specifically, we expect maternal self-efficacy to be negatively associated with anxiety, depression, and stress in adolescent mothers.

H.2 Maternal self-efficacy and emotional well-being in adolescent mothers will be related to the social-emotional development of their children. Notably, we expect higher maternal self-efficacy levels and lower levels of anxiety, depression, and stress in adolescent mothers to be related to better social-emotional development in their children.

H.3 Higher maternal self-efficacy may be associated with fewer problems of emotional well-being in mothers, and thus promote children's social-emotional development.

Therefore, we focused on the following research questions:

1. How are levels of maternal self-efficacy in adolescent mothers related to their emotional well-being?
2. Are the maternal self-efficacy and well-being of adolescent mothers associated in any way with their children's social-emotional development?
3. Does maternal well-being in adolescent mothers mediate maternal self-efficacy and its impact on their children's social-emotional development?

## MATERIALS AND METHODS

### Participants

A non-probabilistic sample of adolescent mothers and their children was recruited from family health centers, hospitals, preschool learning centers and adolescent mothers' residences in the Biobío Region, Chile. Statistical power analyses were previously carried out to determine the required sample size. A statistical power analysis based on Fisher's transformation for the correlation test found that 84 observations were necessary to detect medium-sized correlations. As for the mediation models we followed the results from previous studies (*Fritz & MacKinnon, 2007*) and, given that medium effect sizes were assumed for the paths involved in all indirect effects, a sample of 71 individuals would be adequate to test indirect effects by means of a BCa bootstrap procedure. In all the power analyses, a significance level of 0.05 and a statistical power of 0.8 were established. The following criteria were used for including the dyads of participants in the study: (a) low-income mothers (*i.e.*, household income less than or equal to 678.49 USD for mid-low income and 391.16 USD for low-income, according to the criteria of the Chilean Association of Market Researchers 2019 (*AIM, 2019*)); (b) mothers who became pregnant at 19 years of age or younger; (c) mothers residing in the Biobío Region, Chile; and (d) mothers with children with typical development aged from 10 to 24 months.

The final sample in the current study comprised 79 children, 42 males (53.2%) and 37 females (46.8%), aged from 10 to 24 months (M = 15.5, SD = 4.2), and 79 mothers, aged from 15 to 21 years old (M = 19.1, SD =1.66). The household income (USD) was from 108

**Table 1 Demographic characteristics of adolescent mothers and their children.**

| Childcare support | N | % |
|---|---|---|
| Childcare support provided | 62 | 78.5 |
| Childcare support not provided | 17 | 21.5 |
| **Mother's marital status** | | |
| Single | 34 | 43.0 |
| Boyfriend | 22 | 27.8 |
| Co-habiting with a partner | 23 | 29.2 |
| **Mother's employment** | | |
| Employed | 15 | 19.0 |
| Unemployed | 64 | 81.0 |
| **Mother`s level of education** | | |
| Dropped out of school | 11 | 13.9 |
| Elementary/high school student | 19 | 24.1 |
| Completed high school | 49 | 62.0 |
| **Child gender** | | |
| Male | 42 | 53.2 |
| Female | 37 | 46.8 |
| **Preschool center** | | |
| Attends | 28 | 35.4 |
| Does not attend | 51 | 64.6 |
| **Relatives who live with the child** | | |
| Mother and father | 6 | 7.6 |
| Mother and grandparents | 56 | 70.9 |
| Mother, father, and grandparents | 17 | 21.5 |

to 678 (M = 509.2, SD = 171.3). Table 1 shows descriptive demographic characteristics of adolescent mothers and their children.

## Instruments

### Sociodemographic questionnaire

An ad-hoc sociodemographic questionnaire was used to record the mothers' age, marital status, education level, employment status, income level and whether they were receiving childcare support or help. The same questionnaire was used to record the children's age and gender and whether or not they were attending preschool.

### The parental evaluation scale (Farkas, 2008)

The Spanish version of the Parental Evaluation Scale (EEP) is a self- administered questionnaire used to evaluate levels of satisfaction and self-efficacy feelings regarding motherhood in women with children under 2 years old. The EEP is composed of 10 items scored on an 11-point Likert scale from 0 (strongly disagree) to 10 (strongly agree) to evaluate experiences of motherhood. The scale provides a total score from 0 to 10, which corresponds to the average of the raw scores for the 10 items; a high score indicates high self- efficacy. Internal consistency has demonstrated an adequate reliability of 0.85 for the

total scale, with a range between 0.66 and 0.81 for items (*Farkas-Klein, 2008*). In our sample, Cronbach's alpha value for adolescent mothers was 0.77.

### Hospital anxiety and depression scale (HADS)

The Spanish version of the Hospital Anxiety and Depression Scale (HADS) (*Caro & Ibáñez, 1992*) is a self-administered questionnaire of 14 items that uses a four-point Likert scale of scores from 0 to 3. It was used to evaluate anxiety and depression levels among the mothers who participated in this study. Seven items are used to assess depression, and seven to assess anxiety. Scores for the answers to each item (depression and anxiety) are added together. The scale gives a total score from 0 to 21 for each item; a high score indicates a high level of anxiety or depression (*Zigmond & Snaith, 1983*). Previous research has shown that the HADS is a reliable tool; Cronbach's alpha for anxiety varied from 0.68 to 0.93 (mean 0.83) and for depression from 0.67 to 0.90 (mean 0.82) (*Bjelland et al., 2002*). A factorial analysis has shown an explicit two-factor structure for all groups in the Spanish version, and the results demonstrated the tool's internal consistency and reliability (*Quintana et al., 2003*). With respect to the questionnaire's validity, the correlations with the associated constructs are acceptable or highly acceptable (*Terol-Cantero, Cabrera-Perona & Martín-Aragón, 2015*). In our study, the Cronbach's alpha coefficient was high for anxiety ($\alpha = 0.83$) and moderate for depression ($\alpha = 0.62$).

### Spanish version of parental stress scale (Oronoz, Alonso & Balluerka, 2007)

The Parental Stress Scale (PSS) is a self-administered questionnaire of 12 items scored on a five-point Likert scale ranging from 1 (strongly disagree) to 5 (strongly agree). It evaluates the level at which situations in one's life are perceived as stressful. The items describe feelings and perceptions about the experience of being a parent. The score is obtained from the sum of the answers and ranges from 12 to 60 points. A high score indicates a high level of stress (*Berry & Jones, 1995*). The Spanish version of the PSS has demonstrated good psychometric properties with highly reliable coefficients (internal consistency, $\alpha = 0.81$, and test-retest, $r = 0.73$) and good validity (*Oronoz, Alonso & Balluerka, 2007*). A short, 10-item version of the PSS has shown adequate reliability ($\alpha = 0.82$, test-retest, $r = 0.77$), validity and sensitivity (*Remor, 2006*). A high Cronbach's alpha coefficient was found in this study ($\alpha = 0.85$).

## Ages and Stages Questionnaire: Social-Emotional–Spanish version (Squires et al., 2002)

The Spanish version of the Ages and Stages Questionnaire: Social-Emotional (ASQ-SE) is a parent-completed questionnaire for nine age ranges that is used to evaluate children's social-emotional development from 1 to 72 months. The questionnaire assesses the following areas: self-regulation (the child's ability or willingness to calm or settle down, or to adjust to different physiological or environmental conditions or stimulation); adaptive functioning (the child's ability to cope with bodily needs such as sleeping, eating, toileting and safety); affect (the child's ability to demonstrate their feelings and show empathy for others); social-communication (the child's ability to express their interests, needs, feelings and affective or internal states by interacting with others verbally or

nonverbally); and interaction (the child's ability to respond to others, such as parents and peers). The ASQ-SE questions are rated on a three-point scale to indicate if the child performs a behavior "often or always" (0), "sometimes" (5) or "never or rarely" (10). An additional check box allows respondents to indicate whether the behavior is of concern to the parents; checked concerns score five additional points. The score is assigned according to the answers provided by the caregiver. The score is obtained from the sum of their responses, and lower scores indicate better social-emotional development in children. This tool has demonstrated a Cronbach's alpha ranging from 0.71 to 0.90, with an overall value of 0.84 (*Squires et al., 2002*). In our sample, the reliability was $\alpha = 0.659$.

## Procedure

This research is part of a project of University of Barcelona. Then, ethical approval was obtained from the University of Barcelona's Bioethics Commission (CBUB), the Health Service of Concepción and the Health Service of Biobío, in accordance with the Declaration of Helsinki (1964 and subsequent updates) and the Guidelines for Good Clinical Practice (GCP) established by the World Health Organization (*WHO, 2005*).

Participants were recruited from 50 health centers, five hospitals, 43 preschool learning centers and the National Children's Service in the provinces of Arauco, Biobío and Concepción in Chile. They were contacted by letter, email, and telephone. Seven health centers, one hospital, four preschool learning centers, and one residential home for adolescent mothers (belonging to the National Children's Service) agreed to participate.

The coordinators of the centers were contacted and asked to collaborate in the recruitment of dyads based on the inclusion criteria mentioned above. They were asked to provide a database to communicate directly with families and request their collaboration in the research. Once the families had agreed, we visited them at home and informed them about their participation. Mothers under the age of 18 signed an informed assent form and their legal guardians an informed consent form. Mothers over the age of 18 signed an informed consent form. Attached to this document was an information sheet on confidentiality and the voluntary nature of their participation, with details on their right to withdraw from the study without penalty, as detailed in Law 20.120 on Scientific Research in Human Beings and Law 19.628 on the Protection of Private Life. Lastly, the mothers completed the questionnaires (sociodemographic questionnaire, HADS, EEP, PSS and ASQ-SE).

Finally, the participants were provided with a brief explanation of how to complete the sociodemographic, HADS, EEP and PSS questionnaires when we visited their homes. Before they completed the questionnaires, we read them the answers and asked if they had any queries. Where there were reading comprehension problems, we read them the items in the questionnaires, explained what the item consisted of and completed the questionnaires together. They then answered each questionnaire in approximately 5 min. The ASQ-SE was then administered with the following instruction: "Please read each question carefully and check whether your baby 'often or always,' 'sometimes' or 'rarely or never' performs the behavior indicated. Answer the questions based on your knowledge of your baby's behavior and based on your baby's usual behavior, not your baby's behavior
when he/she is sick, very tired, or hungry. "They completed the ASQ-SE in around 10–15 min. The points for responses to each questionnaire were transferred to a database for subsequent analysis.

### Data analysis

The data were analyzed in various stages. First, the children's social-emotional development and scores for each area were transformed into Z scores to avoid differences in cutoff points according to the age ranges established in the ASQ-SE. With respect to the first objective, that is, to analyze the relationship between maternal self-efficacy and anxiety, depression, and stress levels in adolescent mothers, Pearson's correlations were employed.

Pearson's correlation coefficients were also used to fulfill the second objective, which consisted of analyzing the association between maternal self-efficacy, anxiety, depression, stress and sociodemographic factors in adolescent mothers and the different areas of children's social-emotional development. For categorical sociodemographic factors, mean ASQ-SE scores were compared *via* the Student t-test (for comparing two independent means) or *via* robust ANOVA, followed by *post hoc* Bonferroni (see Table S1).

Multiple mediator models were subsequently estimated to include all relevant variables in one single model. In this regard, six models, one for each response variable corresponding to the different areas of social-emotional development, were estimated based on the regression models described in *Hayes (2012)*. In short, these models allow researchers to estimate direct, indirect, and total effects of the explanatory variable (*i.e.*, mothers' self-efficacy) over the response (*i.e.*, children's social-emotional development). In the case of indirect paths, the effect of the explanatory variable over the response *via* a mediator (*i.e.*, maternal well-being indicators: anxiety, depression, and stress) was estimated by means of the regression procedure mentioned above. To make decisions regarding the estimated effects, two different statistical procedures were used: (a) asymptotic tests in the case of direct effects; and (b) 95% BCa bootstrap CIs with 5,000 replications in the case of indirect and total effects. As a decision rule for the latter, a significant effect was concluded if the non-effect compatible value (*i.e.*, a value of 0) fell outside the boundaries of the estimated CI. PROCESS (*Hayes, 2012*) and IBM SPSS version 27 for windows were used for statistical analyses.

In the current study resampling methods (*e.g.*, BCa-Bootstrap) were used to test mediated as well as total effects instead of asymptotic tests (*i.e.*, Sobel tests), as recommended in previous studies (*Fritz & MacKinnon, 2007*; *Preacher & Hayes, 2004*). Additionally, full models including all paths were taken into account in the BCa bootstrap analyses rather than using other step-based alternatives that are more likely to inflate rates of type I error (such as Baron and Kenny's classical approach).

## RESULTS

Table 2 shows the mean, SD, minimum and maximum scores for maternal factors obtained by the EEP, HADS and PSS and the ASQ-SE raw scores for children's social-emotional development.

**Table 2 Descriptive data on maternal self-efficacy, maternal anxiety and depression, maternal stress, and children's social-emotional development.**

|  | Min. | Max. | M | SD |
|---|---|---|---|---|
| Maternal self-efficacy (0–10) | 3 | 10 | 7.71 | 1.59 |
| Mother's anxiety (0–21) | 0 | 20 | 6.23 | 3.83 |
| Mother's depression (0–21) | 0 | 15 | 5.13 | 3.11 |
| Maternal stress (12–60) | 12 | 45 | 22.18 | 7.68 |
| Social-emotional development total score (0–240) | 0 | 145 | 36.58 | 22.58 |
| Self-regulation (0–60) | 0 | 50 | 12.03 | 10.11 |
| Adaptive functioning (0–40) | 0 | 20 | 5.63 | 5.74 |
| Affect (0–30) | 0 | 15 | 6.58 | 5.58 |
| Social-communication (0–40) | 0 | 20 | 4.18 | 5.80 |
| Interaction (0–70) | 0 | 35 | 6.84 | 7.69 |

Note:
   Raw values of the children's social-emotional development variables.

**Table 3 Correlations between children's social-emotional development and area scores, sociodemographic factors, and maternal self-efficacy, anxiety, depression, and stress.**

|  | 2 | 3 | 4 | 5 | 6 | 7 | 8 | 9 | 10 | 11 | 12 | 13 |
|---|---|---|---|---|---|---|---|---|---|---|---|---|
| 1 Maternal self-efficacy | −0.51** | −0.50** | −0.72** | −0.06 | −0.14 | −0.08 | −0.35** | −0.05 | −0.00 | −0.12 | 0.07 | −0.23* |
| 2 Mother's anxiety |  | 0.62** | 0.45** | 0.03 | 0.06 | 0.09 | 0.20 | 0.03 | 0.07 | −0.08 | −0.07 | 0.14 |
| 3 Mother's depression |  |  | 0.55** | −0.02 | 0.06 | 0.10 | 0.18 | 0.08 | 0.11 | 0.09 | −0.11 | 0.17 |
| 4 Maternal stress |  |  |  | 0.08 | 0.06 | 0.18 | 0.39** | 0.06 | 0.09 | 0.22 | 0.07 | 0.32** |
| 5 Mother's age |  |  |  |  | 0.05 | 0.11 | 0.06 | −0.15 | −0.04 | −0.06 | −0.05 | −0.04 |
| 6 Children's age |  |  |  |  |  | 0.14 | 0.01 | −0.05 | 0.08 | −0.07 | 0.06 | −0.00 |
| 7 Household income |  |  |  |  |  |  | 0.05 | −0.01 | 0.02 | 0.01 | −0.08 | 0.01 |
| 8 Self-regulation |  |  |  |  |  |  |  | 0.27* | 0.36** | 0.24* | 0.15 | 0.82** |
| 9 Adaptive functioning |  |  |  |  |  |  |  |  | 0.07 | 0.09 | −0.17 | 0.43** |
| 10 Affect |  |  |  |  |  |  |  |  |  | 0.13 | 0.16 | 0.52** |
| 11 Social-communication |  |  |  |  |  |  |  |  |  |  | 0.19 | 0.50** |
| 12 Interaction |  |  |  |  |  |  |  |  |  |  |  | 0.43** |
| 13 Social-emotional total score |  |  |  |  |  |  |  |  |  |  |  |  |

Notes:
   * $p < 0.05$.
   ** $p < 0.01$.

    With respect to the relationship between mothers' self-efficacy scores and well-being, maternal self-efficacy was negatively related to maternal anxiety ($r = −0.511$, $p < 0.001$), depression ($r = −0.504$, $p < 0.001$) and stress ($r = −0.715$, $p < 0.001$). This result means that maternal self-efficacy was lower in adolescent mothers who demonstrated higher levels of anxiety, depression, and stress. Regarding the analysis of maternal self-efficacy, anxiety, depression and stress, and their relationship with children's development, bivariate analyses (see Table 3) showed that the total ASQ-SE scores for children's social-emotional development were associated with maternal stress scores ($r = 0.32$, $p = 0.004$) and maternal self-efficacy ($r = −0.23$, $p = 0.046$). Moreover, self-regulation development was related to maternal self-efficacy scores ($r = −0.35$, $p = 0.002$) and

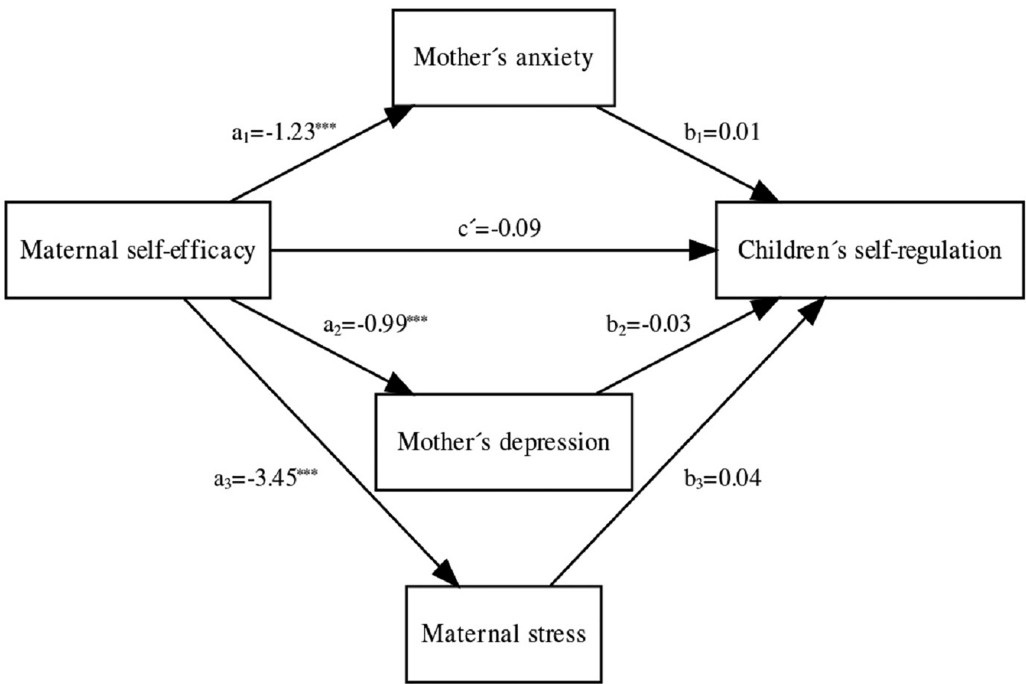

**Figure 1 Asterisks (\*\*\*) indicate the significance of direct effect of maternal self-efficacy on: (a1) anxiety (Coeff. −1.23; $p < .001$); (a2) depression, (Coeff. −99; $p < .001$); and (a3) stress, (Coeff. −3.45; $p < .001$).**

maternal stress scores ($r = 0.39$, $p < 0.001$). These results indicate that children's social-emotional and self-regulation development were better in children whose mothers presented lower stress levels. Furthermore, self-regulation development was better in children whose mothers demonstrated higher maternal self-efficacy. With respect to the relationship between mothers' and children's age, and household income with children's social-emotional development and area scores there were not a significant association between them ($p > 0.05$) (see Table 3).

With respect to the multiple mediation models, the only social development dimension for which there were significant results in terms of predictive capacity was children's self-regulation. For this reason, results for these scales only are presented here, although the other mediation models are included in the Supplemental Materials (see Sections S2 to S6). Figure 1 shows the model that, firstly, specified the direct effects of maternal self-efficacy on the mediating variables: maternal anxiety ($a_1$), maternal depression ($a_2$) and maternal stress ($a_3$). The model also specified the effects of the mediating variables (maternal anxiety, depression, and stress) on children's self-regulation ($b_1$ to $b_3$). All effects were calculated separately, along with the three mediated effects (terms $a_1b_1$, $a_2b_2$, and $a_3b_3$). In Fig. 1, the c' path represents the direct effect of maternal self-efficacy on children's self-regulation while controlling for the mediated effects. Note that in this mediation model, the total effect of maternal self-efficacy on children's self-regulation (*i.e.*, the c term) was thus estimated as the difference between estimates c' and the total mediated or indirect effect ($a_1b_1 + a_2b_2 + a_3b_3$).

**Table 4  Model coefficient effect of maternal self-efficacy on children's self-regulation.**

| | Children's self-regulation | | |
| --- | --- | --- | --- |
| | Coeff. | SE | $p$ |
| Maternal self-efficacy | −0.09 | 0.10 | 0.36 |
| Mother's anxiety | 0.01 | 0.04 | 0.76 |
| Mother's depression | −0.03 | 0.05 | 0.51 |
| Maternal stress | 0.04 | 0.02 | 0.05 |
| Intercept | −0.12 | 1.15 | 0.92 |
| | $R^2 = 0.16$ | | |
| | $F (4, 74) = 3.64$, $p < 0.01$ | | |

The direct effect between maternal self-efficacy and children's self-regulation was nonsignificant ($c' = -0.09$, $p = 0.36$; 95% CI [−0.29 to 0.11]). The indirect effects on children's self-regulation mediated *via* either maternal anxiety ($a_1b_1 = -0.01$; 95% BCa–CI = [−0.11 to 0.08]), maternal depression ($a_2b_2 = 0.03$; 95% BCa–CI = [−0.06 to 0.12]) or maternal stress ($a_1b_1 = -0.14$; 95% BCa–CI = [−0.33 to 0.03]) were not significantly different from zero. However, when all these effects, both direct and indirect, were included in the model, the total effect yielded a significant result ($c = -0.22$; 95% CI = [−0.35 to −0.08]), which indicated multiple partial mediation. This model therefore globally yielded a significant total effect on children's self-regulation, thus indicating that higher maternal self-efficacy results in better child self-regulation and, in turn, reduces the effect of anxiety and stress on self-regulation.

Table 4 shows all estimated arcs in the final model for maternal self-efficacy (X) on children's self-regulation (Y), mediated by maternal anxiety, depression, and stress. This final model with multiple mediation effects accounted for 16% of the variability in children's self-regulation ($F (4, 74) = 3.64$; $p < 0.01$).

## DISCUSSION

The current study aimed to analyze the relationship between maternal self-efficacy and emotional well-being (levels of stress, depression, and anxiety) in adolescent mothers and their association with children's social-emotional development. Maternal well-being was considered as a mediator between maternal self-efficacy and children's social-emotional development. Our results were similar to those of *Vance et al. (2020)*, who found that maternal self-efficacy predicted better maternal psychological well-being and was higher in older mothers. Even though we did not include older mothers to compare maternal self-efficacy levels between different age groups, in the sample of adult Chilean mothers considered to construct the instrument used in this research (*Farkas-Klein, 2008*), the level of maternal self-efficacy was similar to that in our sample. This was an interesting finding because it suggests that maternal self- efficacy, even in young mothers, is closely linked to maternal stress, anxiety, and depression (*Brazeau et al., 2018*; *Seymour et al., 2015*). Thus, strengthening maternal self-efficacy and reducing mothers' emotional distress could be key to ensuring that adolescents have a better experience of motherhood. In light

of the evidence provided by the studies mentioned above, the importance of our results, as pointed out by *Haslam, Pakenham & Smith (2006)*, lies in the fact that high maternal self-efficacy could explain the lower levels of mother's problems of emotional well-being by promoting better self-control, self-confidence and the use of strategies to face challenges. In Chile, some studies have indicated that intervention programs for adolescents during the early stages of motherhood contribute to better maternal mental health (*Aracena et al., 2009*, *2011*). In this regard, our findings justify the evaluation of mothers' anxiety, depression, and stress levels, in addition to key information regarding the close relationship between maternal self- efficacy and psychological well-being. This contributes to mothers' sense of ability to tackle behavioral problems and demanding situations involving their children (*Hastings & Brown, 2002*).

Moreover, our findings showed that some maternal factors were related to children's social-emotional development. Specifically, a relationship was identified between maternal stress, maternal self-efficacy, and children's social-emotional development. The results concerning the association between maternal stress and social-emotional development are consistent with the findings of *Palmer et al. (2018)*, who demonstrated that children's social-emotional development problems were related to higher levels of stress reported by healthy adolescent and adult mothers who were followed up from pregnancy until their children were 2 years old. *Pesonen et al. (2008)*, in particular, indicated that maternal stress interferes in children's self-regulation, probably because the stability of maternal stress is related to the child's emotional regulation (*Williford, Calkins & Keane, 2007*). Likewise, as mentioned in the introduction, social-emotional development, which could include a disruptive temperament and more externalizing behavioral problems in children, is related to higher maternal stress levels (*Gutteling et al., 2005*). The above mentioned findings might explain the significant relationship between higher maternal stress levels and negative children's self-regulation observed in the present study. In this regard, the sample employed in this study comprised low-income mothers, as with the sample used by *Bush et al. (2017)*, who examined the association between maternal stress and children's self-regulation from gestation to the postpartum period. In this latter study, mothers who perceived themselves as more stressed during pregnancy and after childbirth reported that their babies had lower levels of self-regulation.

The studies mentioned considered mothers' perceived stress. They included questionnaires to assess children's social-emotional development, as in our research, which measured children's social-emotional development with the ASQ-SE through a questionnaire completed by adolescent mothers. *Salomonsson & Slued (2010)*, who studied a sample of mothers suffering from emotional problems, observed a relationship between social-emotional development levels measured with the ASQ-SE and maternal stress levels evaluated through a questionnaire, and therefore concluded that the ASQ-SE could be used as a tool to anticipate mental health problems in mothers, rather than just child development problems. Unlike this study, our sample did not consist primarily of mothers with high maternal stress levels. Despite this, our findings contribute to the literature that explains how maternal stress could influence children's social-emotional

development in low-income young mothers, as occurs in samples of older mothers and mothers with mental health problems.

Concerning the association between maternal self-efficacy and children's social-emotional development, our results were consistent with a study by *Treat et al. (2020)*, who found that maternal self-efficacy was positively related to better social and emotional development in children in a sample of low-income mothers. They also suggested that mothers who receive social support have higher levels of parental self-efficacy, since this could contribute to greater self-confidence in parenting skills. This is an important issue, given that maternal self-efficacy can vary depending on the support the mother receives from family, friends and the child's father, as well as her emotional well-being (*Coleman & Karraker, 1998*, *2003*). However, this variable was not considered in our research, but it is relevant information that could be considered in future studies on young mothers, since an adolescent mother who receives social support may have higher levels of maternal self-efficacy. *Shahry et al. (2016)* demonstrated that women who had unintended pregnancies perceived that their levels of social support and self-efficacy were lower, and that low maternal self-efficacy could generate frustrating feelings related to parenting. In a sample of dyads between low-income mothers and infants, *Bates et al. (2020)* found an association between higher levels of maternal self-efficacy and fewer problems of self-regulation in the children. The instrument used in that study measured aspects related to the child's ability to be calm, like the instrument used in our study. According to the authors, infant care that included parents' feelings, behaviors, and thoughts influenced the dyadic relationship between mother and child.

Our study did not find a significant association between maternal self-efficacy and well-being and children's adaptive functioning, or any of the other ASQ-SE areas, unlike the studies by *Micklewright et al. (2012)* and *Stika et al. (2015)*. However, the results of research carried out by *Voliovitch et al. (2021)* were similar to ours. These authors showed that the parenting stress associated with low socioeconomic status was not related to adaptive functioning in children at risk of autism spectrum disorder. Even though the study sample comprised children at risk of autism spectrum disorder, this was not related to parental stress. The discordance between maternal stress and children's adaptive functioning (which focuses on the bodily needs of children, such as eating and sleeping) may be due to different biological factors, such as breastfeeding, which has been shown to be associated with a reduction in infant feeding problems (*Schmid et al., 2011*), smoking in breastfeeding mothers, which affects infant sleep (*Mennella, Yourshaw & Morgan, 2007*), and parents' knowledge of and adaptation to their children's sleeping and feeding patterns (*Rosen, 2008*), variables that were not evaluated in this research. In terms of the other areas measured, social-communication, interaction and affect were not found to be related to maternal factors (maternal self-efficacy, anxiety, depression, and stress). Although we already know that maternal self-efficacy and the mother's emotional well-being play a prominent role in children's social-emotional development, most previous studies have related them to attachment security (*Groh et al., 2017*; *O'Connor, Collins & Supplee, 2012*). This aspect should be considered in future studies on dyads of adolescent mothers and children's social-emotional development.

Regarding maternal emotional well-being as a mediator of maternal self-efficacy and children's social-emotional development, our results partially differ from those of *Weaver et al. (2008)*, who found that maternal depression mediates the relationship between parental self- efficacy and childhood behavioral problems. According to our results, children's self-regulation was better when maternal self-efficacy was higher if the effect of maternal emotional distress decreased, since the combined total model we used for this research was significant. This means that children's self-regulation is better when maternal self-efficacy is higher if the effect of anxiety and stress is reduced, but the inverse effect occurred with depression. The indirect effects of maternal self-efficacy, mediated by anxiety, depression, and stress, on children's self-regulation were nonsignificant in our sample. This result could be due to the small sample size and the fact that a low number of adolescent mothers had high levels of depression. Furthermore, the low rates of maternal depression could also explain why the children in our sample did not show high scores for problems of social-emotional development—a finding that is at odds with previous reports in the literature that the children of adolescent mothers have more significant behavioral problems, lower levels of social performance and poorer well-being (*Hofferth & Reid, 2002*; *Levine, Pollack & Comfort, 2001*). Despite this, our results are similar to the findings of *Van den Heuvel et al. (2015)*, who demonstrated that the presence of maternal anxiety could be a mediating factor between maternal mindfulness, which is associated with optimism and feelings of competence, and self-regulation problems in infants. In light of the previous studies presented here, the importance of the self-confidence of mothers as caregivers of their children has relevant implications for their emotional well-being and their children's development. However, this aspect should be considered in future studies on adolescent mothers with higher levels of emotional distress. From this standpoint, this study may contribute to the development of theoretical models to explain the effect of specific maternal factors, which could then be tested with larger samples and longitudinal research. However, the mediation models using other ASQ-SE areas (adaptive functioning, affect, social communication, and interaction) and the total score for social-emotional development presented a low predictive capacity that could not be considered useful. These results could be explained by the fact that this study showed no significant correlation between maternal factors (*i.e.*, maternal self-efficacy, depression, anxiety, and stress) and the rest of the areas of social-emotional development. Furthermore, most of the studies which have found an association between maternal self-efficacy and emotional well-being and children's social-emotional development have been carried out with adult mothers rather than adolescents (*Treat et al., 2020*). Likewise, studies of mothers' emotional well-being problems focused on the perinatal (*Junge et al., 2017*) and postnatal (*Letourneau et al., 2012*) periods, but did not consider whether these problems originated at these stages, and in any case in our sample the children were older. Another aspect to consider is that social support in motherhood plays a key role in maternal self-efficacy and emotional well-being, as pointed out by *Haslam, Pakenham & Smith (2006)*, who measured how mothers perceived social support. Although in our study we asked adolescent mothers if they received support in childcare, we did not inquire about the quality of the support or how they perceived it. Their

perceptions of these issues may well have influenced the maternal factors studied and may have interfered in their children's social-emotional development.

Despite this, the results obtained confirm the importance of the emotional well-being of adolescent mothers in terms of its influence on the social-emotional development of their children, as mentioned in the introduction. Previous studies on Chilean adolescent mothers have shown that those with better emotional stability have children with a lower probability of developmental delay (*Aracena et al., 2011*; *Olhaberry et al., 2015*). However, few studies have analyzed this issue in depth in Chile. Therefore, considering the importance of the results of our study, it is necessary to explore how emotional state and self-perception influence the role played by adolescent mothers in the development of their children. Thus, our results support the need to include adolescent mothers in research related to emotional well-being (*Leerlooijer et al., 2014*; *Marino et al., 2016*; *Meadows-Oliver et al., 2007*), maternal self-efficacy (*Ford et al., 2005*) and children's development in light of their environment (*Keown, Woodward & Field, 2001*). In the Chilean context in particular, adolescent mothers could be considered an at-risk population, since there is a significant association between low socioeconomic level and adolescent pregnancy.

This population has been subject to growing inequality in recent years, with a higher pregnancy rate in low-income adolescents; moreover, even though the adolescent pregnancy rate has decreased in Chile, the intergenerational transmission of poverty in adolescent mothers continues to be a problem (*Lavanderos et al., 2019*). It should also be noted that *Farkas & Valdés (2010)* carried out research into Chilean low-income mothers and found that older mothers presented lower levels of stress related to the child's characteristics and that, in turn, increased maternal stress leads to a reduction in perceived self-efficacy regarding motherhood. Our findings suggest, as proposed by *Boyce et al. (2017)*, that strengthening parental self-efficacy would help improve emotional well-being levels and children's social-emotional development. From this standpoint, studies that take this approach are essential to draw conclusions on how the emotional well-being and maternal self-efficacy of Chilean adolescent mothers and their families are associated with their children's development and behavior.

Other studies in samples of Latin American women have referred to the issue of economic and educational inequality associated with early motherhood (*Marteleto & Villanueva, 2018*; *Rodríguez-Vignoli & Cavenaghi, 2014*) and have assessed the impact of the context on their children's social-emotional development (*McDermott et al., 2021*). Although one of the more prosperous countries in Latin America (*Banco Mundial, 2020*), Chile does not differ greatly from other countries in the continent in terms of adolescent mothers' characteristics, and so the results presented here can perhaps be considered as a reference point for research of this nature in other countries.

The present study contributes to the literature on the importance of emotional well-being and maternal self-efficacy on the social-emotional development of children of adolescent mothers. However, the study presented several limitations that should be considered when interpreting the results. First, the study sample size was moderate, which may have limited the statistical power when testing conjecture effects. Nevertheless, as

reported in a literature survey by *Fritz & MacKinnon (2007)*, this is a common problem in psychology; 40% of the studies reported in their survey used moderate to low sample sizes (below 150 observations). In order to mitigate the problem of the sample size, the current study is based on the use of a bias-corrected bootstrap confidence interval, as this is considered a trustworthy test (*Hayes & Scharkow, 2013*). More research is needed to confirm our results. Additionally, the correlational design did not permit us to establish a clear causality, so it would be necessary to treat the "effect" concept used in the mediation analysis with caution. The "effect" reported here was estimated based solely on the covariation between children's self-regulation, maternal self-efficacy and maternal emotional well-being scores. In this regard, covariation is a condition that does not necessarily imply causality. The present study used a cross-sectional design, which clearly limited our ability to draw causal inferences regarding the relationships between the variables. However, the proposed models were supported by previous research and by current theory, and all the relationships estimated between the mothers' self-efficacy and well-being, and children's social development have also been demonstrated by longitudinal studies (*e.g.*, *Huang et al., 2014*; *Lawrence et al., 2020*; *Van den Heuvel et al., 2015*; *Vance et al., 2020*). Other studies of these relationships have also applied cross-sectional designs (*e.g.*, *Azmoude, Jafarnejade & Mazlom, 2015*; *Puspasari, Nur Rachmawati & Budiati, 2018*; *Tan et al., 2020*). In any case, the findings of the present study would need to be replicated by a longitudinal design to establish temporal order between the exogen and endogen variables, so that the mediation effects can be adequately tested.

Another limitation to consider is that all tools used to analyze the data were questionnaires. Therefore, the adolescent mothers' self-assessment of maternal self-efficacy and emotional well-being and their completion of questionnaires on their children's social-emotional development may have generated biases that could have interfered with our results. Furthermore, as in the study by *Krijnen, Verhoeven & van Baar (2021)*, child development measured with ASQ-SE was reported by the mother. It is well known that mothers' emotional well-being problems can affect their perceptions of their children's development or behavior (*Closa-Monasterolo et al., 2017*); therefore, descriptions of their children's development may be based on their emotional state. Likewise, Chilean low-income adolescents have lower linguistic skills than those who from higher economic sectors (*Agencia de Calidad de la Educación, 2018*), and so there is a risk that the questions were not fully understood. From this perspective, the significant associations may be due to a functional relationship, but it is also possible that they are due to individual and subjective tendencies of the mother answering the questionnaires.

Despite the limitations of the design, according to *Rothbart & Bates (2007)*, questionnaires completed by parents to measure child development can be advantageous because parents have observed their children in different situations for longer than external observers. Their responses may therefore be more accurate. For future research, we suggest that questionnaires on children's social-emotional development questionnaires and behavioral observation measures be complemented.

Another aspect to consider is that the study did not take account of the parental self-efficacy levels and emotional well-being of fathers involved in childcare. Therefore, the results may have varied depending on family structure, especially where the father was present (*Fagan et al., 2014*; *Cabrera et al., 2008*). However, although the father's presence is important, not all adolescent mothers are supported by the father of their children. In the Chilean context, several adolescent fathers indicated that they did not live with the mother of their child, nor with their child, and that the relationship was unstable with a low level of commitment, which could lead to the breakdown of the couple's relationship and risk their closeness or even their contact and affectional bond with their children (*Molina-Gutiérrez, 2011*). Future studies on Chilean adolescent mothers could consider a sample in which the fathers fulfill their parenting responsibilities. The intention would be to analyze fathers' parental self-efficacy and emotional well-being, and the adolescent mothers' degree of satisfaction with the fathers' involvement in the care of their children, as in a study carried out by *Fagan & Lee (2010)* with a sample that included, among other ethnicities, Latina adolescent mothers. Further analyses of adolescent mothers and their children are needed to determine the consistency of our findings.

## CONCLUSIONS

Overall, the present study confirmed that maternal self-efficacy could explain lower levels of emotional well-being problems experienced by adolescent mothers. In turn, when the emotional well-being of mothers mediates maternal self-efficacy, this could be a determining factor for their children's social-emotional development. However, future studies should use larger samples and perform further analyses, (for instance, of sociodemographic variables and their possible relationship with children's social-emotional development). The above findings are necessary to explore how the emotional and self-perceived state influences the role of mothers in the development of their children, especially in adolescent mothers. In this context, maternal factors should be considered when planning interventions to prevent childhood behavioral problems. These dyads, therefore, form part of a group that should receive significant attention in Chilean national programs. These findings highlight the importance of providing adolescent mothers with social support and guidance during pregnancy onwards and intervention programs tailored to their emotional needs. A growing understanding of the maternal factors analyzed should help evaluate the research results associated with this topic and the effectiveness of intervention efforts to reduce negative family implications that could generate problems in the social-emotional development of children born to Chilean adolescent mothers.

## ACKNOWLEDGEMENTS

The authors would like to thank all families who agreed to participate in this research and the workers at the health centers, preschool learning centers, the hospital and the residential home for adolescent mothers that made a substantial contribution to this work.

### Funding

This study was supported by the Agencia Nacional de Investigación y Desarrollo (ANID) through a doctoral fellowship granted to Laura Léniz-Maturana (reference number: 72190318). The funders had no role in study design, data collection and analysis, decision to publish, or preparation of the manuscript.

### Grant Disclosures

The following grant information was disclosed by the authors:
This study was supported by the Agencia Nacional de Investigación y Desarrollo (ANID) through a doctoral fellowship granted to Laura Léniz-Maturana (reference number: 72190318). The funders had no role in study design, data collection and analysis, decision to publish, or preparation of the manuscript.

### Competing Interests

The authors declare that they have no competing interests.

### Author Contributions

- Laura Léniz-Maturana conceived and designed the experiments, performed the experiments, analyzed the data, prepared figures and/or tables, authored or reviewed drafts of the paper, and approved the final draft.
- Rosa Vilaseca conceived and designed the experiments, performed the experiments, authored or reviewed drafts of the paper, and approved the final draft.
- David Leiva analyzed the data, prepared figures and/or tables, authored or reviewed drafts of the paper, and approved the final draft.

### Human Ethics

The following information was supplied relating to ethical approvals (*i.e.*, approving body and any reference numbers):

Ethical approval was obtained from the University of Barcelona's Bioethics Commission (CBUB), the Health Service of Concepción and the Health Service of Biobío.

### Data Availability

The raw data is available in the Supplemental File.

### Supplemental Information

Supplemental information for this article can be found online at http://dx.doi.org/10.7717/peerj.13162#supplemental-information.

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
