# Peer review of "Maternal self-efficacy and emotional well-being in Chilean adolescent mothers: the relationship with their children’s social-emotional development"

_PeerJ, doi:10.7717/peerj.13162_

## Round 0.1 · original submission · Major Revisions

Please authors, kindly find the reviewers' comments on your work, which I believe will improve it. Kindly attend to them in the very best detail. Thank you

Reviewer 1 ·

Basic reporting

Dear Editor,

Thank you for considering me as a reviewer for this publication in your journal PeerJ.

The manuscript “Maternal self-efficacy and emotional well-being in Chilean adolescent mothers: the relationship with their children’s social-emotional development” by Léniz-Maturana et al. deals with the associations between maternal self-efficacy, maternal well-being, and children’s socio-emotional development in a selected sample of low-income adolescent mothers in Chile. The authors underline the importance of maternal self-efficacy and maternal well-being for an optimal socio-emotional development of the child.
This is a nicely written, interesting and relevant manuscript. Furthermore, this is a valuable contribution to science and clinical practice and I suggest acceptance for publication (with revisions).

The study has several strengths. These include the importance of the questions the authors are addressing for the field as well as the South American sample because I like the idea of cultural diversity in research.

1. BASIC REPORTING
- Clear, unambiguous, professional English language used throughout.
Please revise the following:

There are multiple, long sentences paragraphs throughout your manuscript, which reduces the readability of your paper. Please work to shorten these sentences, e. g. lns. 58-62; lns. 95-98; lns. 490-494; lns. 531-535

Please check the punctuation throughout the manuscript: commas are missing, e. g. ln. 142: “social isolation, and low self-esteem; ln.155; ln. 419; ln. 430; ln. 442; ln. 15: “anxiety, depression, and stress” (according to the most recent publication manual of the American Psychological Association, I believe the 7th edition).

You could make it easier for the reader to follow; please revise the following sections: sentence lns. 75-77; ln. 148: what do you mean with “childhood behavior”?; ln. 152: “complicated stage”; ln. 160: “development process is incomplete”

Ln. 56: “fortunately” – please avoid personnel opinion

- Intro & background to show context. Literature well referenced & relevant.
Please revise the following:

Why is Chile the focus of this study? What is unique about Chile? These facts are important to understand and contextualize the results, and then to discuss whether they would be applicable to other contexts. You mentioned some points (from ln. 50) but it would help if the rationale was set out more explicitly.

Please insert sources for the following statements: lns: 68-69; lns: 106-108; lns: 131-132; lns: 134-135; lns: 179-181; lns: 190-191; lns: 206-207

Lns.: 94-95: why you used the word “consequently”? I am sorry; I did not understand the link.

Lns: 147-148: Somewhat the last sentence comes in as a surprise. You might revise it, so that the reader can follow your reasoning.

Lns. 161-166: Please try to streamline the arguments, sometimes I do not get the point; also lns. 190-200.

Lns. 201-203: “different studies” – please name more than one reference.

Ln. 246; ln. 256; ln. 271; ln. 282: please add the references of the scales and inventories.

- Structure conforms to PeerJ standards, discipline norm, or improved for clarity. - YES

- Figures are relevant, high quality, well labelled & described.
Please revise the following:

Figure 1: the figure is very helpful; unfortunately, the resolution is not good enough.

- Raw data supplied (see PeerJ policy). - YES

Experimental design

2. EXPERIMENTAL DESIGN
- Original primary research within Scope of the journal. - YES

- Research question well defined, relevant & meaningful. It is stated how the research fills an identified knowledge gap.
Please revise the following:

The presentation of the relevance of the sample (why chile? Why the selected sample of low-income adolescent mothers?) and the introduced concepts could be made clearer in the introduction.

Ln. 213: Hypothesis C: maternal self-efficacy is named the mediator – please clarify.

Ln. 249: from 0 to 10 -> 11point likert scale

- Rigorous investigation performed to a high technical & ethical standard. - YES

- Methods described with sufficient detail & information to replicate. - YES

Validity of the findings

3. VALIDITY OF THE FINDINGS
- All underlying data have been provided; they are robust, statistically sound, & controlled.
Please revise the following:

Please give information about the correlations (or t-tests) of the study variables and the sociodemographic (control) variables, e. g. age of mother and child, income, level of education, support, etc. - this should be presented in the Results section. The role of control variables should be discussed as a future direction (potential moderators, explanatory as to why some expected associations may not exist - e.g., maternal depression and child development)

Table 1: I find it hard to read, please change the design.

Table 2: Please report the raw values of the children’s socio-emotional development variables and – for a better overview – please add also the means and SDs of the maternal variables (and delete the passage in the text). For example in Lns. 201-203 you write: “Different studies agree that the children of adolescent mothers develop more slowly than those of non-adolescent mothers, and one of the possible causes of these outcomes is the socially disadvantaged origin of adolescent mothers.” It would be interested to know how the children in your sample are developed (e.g. with reporting the raw scores).

It is not until line 229 that the reader learns that this is a selective sample of low- income mothers. Why? Please give that information already in the abstract.

Lns. 487-489: An additional avenue for investigation (and perhaps clarification) might be determining the item-level bivariate associations between these measures with your data.

Ln. 363: self-efficacy mean of 7.71 is relatively high, or not? In this selected sample: adolescent and low-income? Please discuss this result further.

E.g. ln. 371: please write the letters r and p in italics; and please revise this throughout the manuscript

Ln. 371: A line break should not be in a parenthesis, same ln. 414.

Ln. 377: please add the exact p value, if p is higher than .001; same ln. 378; ln. 379; ln. 405; e.g. “p = .034”

- Speculation is welcome, but should be identified as such.
Please revise the following:

The authors sometimes use a causal language when discussing the results. For example, on lns. 221-223 “protective factor”; lns. 430-433 “effects”, ln. 436 “impact”. I urge the authors to be cautious when using this language and to substitute causal terms such as “impact” with more appropriate ones such as “association”. Thus, there are reasons to be cautious in strong statements about mediation.

- Conclusions are well stated, linked to original research question & limited to supporting results.
Please revise the following:

Lns. 539-541: – many of the relations were insignificant, mother’s anxiety and depression scores did not correlate with the socio-emotional development of the child; you discussed this null finding but, in my opinion, you should make it more clear; same in line 615.

The relevance of intervention was nicely displayed.

Additional comments

There are a number of limitations that need to be acknowledged and discussed. For example:

Please add as a further limitation: Generalization of the results is limited because selected sample of low-income adolescent mothers.

A further issue is that all data stem from the report of one person, i.e. mother, who rated both her own feelings and the behavior of the child. Thus, associations found between measures may be methodological artefacts. In the case of a significant association we do not know whether there is really a functional relationship between them or whether this is due to individual tendencies of the mothers to describe own behavior and child behavior, general tendencies to respond more or less positive when filling in questionnaires. For example, it may be that some mothers have a more pessimistic (as compared to others with a more positive) view of life and, therefore, they tend to describe both their own emotional state and the child's emotional problems.

References: Lns. 895-898: why “S” bevor “Q”?

·

Basic reporting

No comments

Experimental design

Dear
please explain mothers that entry your study were nullipara? if yes why did you collected sampling from preschool.
2- How did you estimate sampling? 79 person is very low for modeling method.
3- How did you calculate mediating role of anxiety and stress?
4- Did you use structural equation model?
5- which software use for assessment of model?
6-which approach Baron &Kenny or Sobel criterion are used for mediating role assessment?
7- please explain about conceptual model in introduction?
8- in structural model only one model is assessed? why did you write five model.
please concentrate only in one model .
9- Please each hypothesis is mentioned in end of introduction.
10-samplig size estimation, references and formula are added in methods.
11.Ethical code is mentioned in paper.
12. Please explain about sampling method? How did you do sampling?

Validity of the findings

No comments

Additional comments

No comments.

Reviewer 3 ·

Basic reporting

Literature references, sufficient field background/context provided

The clear and detailed way by which authors described the theretical framwork of the study and previous studies is appreciale. However, my suggestion is to reference more recent studies both for what concern parental self-efficacy beliefs (e.g. Glatz & Buchanan, 2015), and for what concerns maternal self-efficacy in adolescent mothers.

Professional article structure, figures, tables. Raw data shared

The article is well articulated and clear in its structure. However, I think that some Table are pretty difficult to read, and therefore, that is necessary a reformattation of them. In particular, Table 4 is difficult to understand. In addition, even if Table 2 shoud report the mean levels of the variables of interest, it reports only the minimum and maximus levels of these.

Experimental design

No comments

Validity of the findings

No comments

Additional comments

ABSTRACT
#1. Line 23. I suggest to the authors to report in the abstract section the mean age of the mothers.

INTRODUCTION
#2. Line 77. I suggest to the authors to report in the introductory section the overall im of the study and its research questions.

The concept of maternal self-efficacy
#3. Line 83. Regarding the definition of parental self.efficacy, I suggest to use the definition prosposed by Bandura (1997).

The relationship between maternal self-efficacy and well-being during adolescence andchildren’s social-emotional development
#4. Line 210. Please, better describing the hypotheses nad the expected results in term of findins from previous studies or theories.

Instruments
#5. I suggest, fir each instruments, to provide an item example.

Data analyses
#6. I was wondering what type of statistical software has been used ti implement the statistical analyses and the reasons why authors did no implemented a structural equation model to examine the mediation.
#7. I suggest to the authors to keep under control the effects of some covriates in the mediation model, such as, participants age and child's gender.

---

## Round 0.2 · accepted · Accept

Thank you authors for the thorough revision that was carried out. Both reviewer and I are very satisfied with the current version, and are convinced it is acceptable for publication. Thanks to the peer review process, the authors improved the quality of the work. Thank you authors for finding PeerJ as your journal of choice. Looking forward to your future scholarly contributions.

Congratulations and best wishes

Reviewer 3 ·

Basic reporting

No comment. The article is clear in each sections, the English is professional and the literature references cover the study's background and objectives.

Experimental design

Research questions are innovative and well defined, also the methods and the procedure fit very well with the manuscript's objectives.

Validity of the findings

Study's findings are very interesting and they have a high impact both in term of practical implications and of theoretical knowledge.

Additional comments

The revisions made to the article address each points raised by reviewers.
The article is now much improved.